

# Dramatic impact of future climate change on the genetic diversity and distribution of ecologically relevant Western Mediterranean *Carex* (Cyperaceae)

Carmen Benítez-Benítez[1], María Sanz-Arnal[2], Malvina Urbani[3], Pedro Jiménez-Mejías[2] and Santiago Martín-Bravo[1]

[1] Department of Molecular Biology and Biochemical Engineering/Botany Area, Universidad Pablo de Olavide, Seville, Seville, Spain
[2] Department of Biology, Universidad Autónoma de Madrid, Campus Cantoblanco, Madrid, Madrid, Spain
[3] Department of Chemistry and Pharmacy, University of Sassari, Sassari, Sardinia, Italy

Corresponding author
Carmen Benítez-Benítez,
cbenben1@upo.es

## ABSTRACT

Anticipating the evolutionary responses of species to ongoing climate change is essential to propose effective management and conservation measures. The Western Mediterranean Basin constitutes one of the hotspots of biodiversity where the effects of climate change are expected to be more dramatic. Plant species with ecological relevance constitute ideal models to evaluate and predict the impact of climate change on ecosystems. Here we investigate these impacts through the spatio-temporal comparison of genetic diversity/structure (AFLPs), potential distribution under different future scenarios of climate change, and ecological space in two Western Mediterranean sister species of genus *Carex*. Both species are ecologically key in their riparian habitats, but display contrasting distribution patterns, with one widespread in the Iberian Peninsula and North Africa (*C. reuteriana*), while the other (*C. panormitana*) is a restricted, probably endangered, Central Mediterranean endemic. At present, we found a strong genetic structure driven by geography in both species, and lower values of genetic diversity and a narrower ecological space in *C. panormitana* than in *C. reuteriana*, while the allelic rarity was higher in the former than in *C. reuteriana* subspecies. Future projections predict an overall dramatic reduction of suitable areas for both species under all climate change scenarios, which could be almost total for *C. panormitana*. In addition, gene diversity was inferred to decrease in all taxa, with genetic structure reinforcing in *C. reuteriana* by the loss of admixture among populations. Our findings stress the need for a reassessment of *C. panormitana* conservation status under IUCN Red List criteria and the implementation of conservation measures.

## INTRODUCTION

Nowadays, anthropogenically-driven global climate change (GCC herein) is one of the main threats to life on our planet. The main conclusion from the Intergovernmental Panel on Climate Change (*IPCC, 2014*) is that contemporary warming of land and ocean is irrefutably taking place, and since the 1980s each successive decade has been warmer than any preceding one. Projections of change in global mean temperature for the period 2016–2035 point to an increase of 0.3–0.7 °C compared to the 1986–2005 period. By the end of this century (2081–2100), estimates predict a higher frequency of extreme weather events and an increase of mean global surface temperature of 0.3–4.8 °C, depending on the emission and climate variability scenario considered (*IPCC, 2013*, *2014*).

Many studies confirm that GCC has already affected the biota and ecosystems (*Parmesan, 2006*; *Johnson et al., 2017*; *Pecl et al., 2017*; *Ripple et al., 2017*; *García & Jordano, 2021*). The responses of species populations to climate change can be categorized into three main types: migration, extinction, and *in situ* adaptation (*Christmas, Breed & Lowe, 2016*). Specifically, ongoing GCC is inducing changes in the distribution of plant species with greater capacity for migration to areas of higher latitude and/or altitude that may provide suitable environmental conditions (niche conservatism, *e.g.*, *Jump & Peñuelas, 2005*; *Wiens et al., 2010*; *Morueta-Holme et al., 2015*; *Christmas, Breed & Lowe, 2016*). However, migration rates for most species could be insufficient to match the pace and magnitude of predicted climate changes (*e.g.*, *Loarie et al., 2009*; *Freeman & Freeman, 2014*; *Morueta-Holme et al., 2015*; *González-Varo, López-Bao & Guitián, 2017*), leaving *in situ* adaptation as the only alternative to extinction for these species. In species with restricted ranges, low genetic variability and/or narrow ecological tolerance, genetic and phenotypic variation may not provide enough adaptability to the new local conditions (*Jump & Peñuelas, 2005*; *Aitken et al., 2008*; *Aubin et al., 2016*; *Christmas, Breed & Lowe, 2016*; *Gray, 2018*).

Species distribution models (SDM, *Guisan & Thuiller, 2005*) are useful tools to predict and compare present and future potential distribution of species (*e.g.*, *Blanco-Pastor, Fernández-Mazuecos & Vargas, 2013*; *Velásquez-Tibatá, Salaman & Graham, 2013*). This helps to assess species vulnerability under different GCC future scenarios (*e.g.*, *Beaumont, Hughes & Pitman, 2008*; *Forester, DeChaine & Bunn, 2013*). While these methods do not take into account population evolutionary potential, which may enable *in situ* adaptation (*Soberon & Peterson, 2005*; *Razgour et al., 2019*), they are still useful to evaluate GCC impact on extant populations under hypotheses of niche conservatism (*Ackerly, 2003*; *Wiens & Graham, 2005*), which is the most common evolutionary response of lineages (*Liu et al., 2012*; *Lososová et al., 2020*; *Sanz-Arnal et al., 2021*). They do not either consider habitat destruction and fragmentation directly produced by human action. Furthermore, several studies have projected current genetic diversity to future times, assuming the extinction of populations falling outside the inferred potential distribution (*Espíndola et al., 2012*; *Lima et al., 2017*; *Rizvanovic et al., 2019*), while others have projected future genetic structure based on molecular and climatic data (*Jay, 2012*; *Jay et al., 2015*).

 

The Mediterranean Basin constitutes one of the Earth's hotspots of biodiversity and harbors an exceptional plant diversity, featuring a high level of endemicity (*Myers et al., 2000*; *Mittermeier et al., 2011*; *Vargas, 2020*). Unfortunately, this will likely be one of the regions most affected by GCC (*Fletcher & Zielhofer, 2013*; *IPCC, 2013*; *Guiot & Cramer, 2016*; *Lionello & Scarascia, 2018*; *Cramer et al., 2018*; *Fenu et al., 2020*; *García & Jordano, 2021*). As a result of this, about 30% of the Mediterranean Basin hotspot could be losing its current climate and turning into a non-analogous one, which could threaten about 30% of its endemic species, assuming loss of analogue climate as a proxy for habitat loss (*Bellard et al., 2014*). Specifically, the Western Mediterranean includes numerous areas that have been considered climatic refugia during historical climatic oscillations from Miocene to Pleistocene (*Médail & Diadema, 2009*). These refugia are reservoirs of unique genetic diversity and evolutionary potential due to the long-term persistence of species, and therefore have high conservation priority (*Médail & Quezel, 1999*; *Médail & Diadema, 2009*).

*Carex* L. (Cyperaceae) is a megadiverse angiosperm genus with several characteristics that makes it an ideal model for the study of the effects of GCC on its species, populations and in the ecosystems where it lives. With c. 2,000 species it ranks among the three largest angiosperm genera in the world (*POWO, 2020*; *Roalson et al., 2021*). It has an almost cosmopolitan distribution although higher species diversity in temperate and cold areas of both hemispheres. *Carex* diversification seems to have been historically favoured by global cooling periods (*Martín-Bravo et al., 2019*), which lead to think that GCC could be negative for its species at least from an evolutionary perspective. In the Western Mediterranean, as in many other world regions, *Carex* species dominate a variety of plant communities, from wetland and river shores to peat bogs and high mountain meadows. Some of these habitats are considered of special interest and have conservation priority in the Directive 92/43/CEE (https://eur-lex.europa.eu/legal-content/ES/TXT/?uri=celex%3A31992L0043). Therefore, responses of ecologically important *Carex* species to climate changes could be probably decisive for the future persistence of these habitats.

*Carex reuteriana* Boiss. and *C. panormitana* Guss. (sect. *Phacocystis*), are two endemic sister species disjunctly distributed in the Western Mediterranean Basin (*Luceño & Jiménez-Mejías, 2008*; *Jiménez-Mejías et al., 2011*), growing in creeks and river shores at medium altitudes (*Benítez-Benítez et al., 2018*). *Carex reuteriana* comprises two subspecies: *C. reuteriana* subsp. *reuteriana* from the centre to NW Iberian Peninsula and *C. reuteriana* subsp. *mauritanica* (Boiss. & Reut.) Jim.-Mejías & Luceño from NW Africa (Morocco) to the S Iberian Peninsula with populations on both sides of the Guadalquivir Valley (Sierra Morena and Betic ranges; *Jiménez-Mejías et al., 2011*). The conservation status of *C. reuteriana* has not been evaluated, although it is seemingly not endangered due to its relatively large distribution range and high numbers of populations and individuals in many of them. On the other hand, *C. panormitana* is a restricted Tyrrhenian endemic from Sardinia with a few additional small populations in Sicily and Tunisia (*Pignatti, 1982*; *Urbani, Gianguzzi & Ilardi, 1995*; *Gianguzzi et al., 2013*; *Jiménez-Mejías et al., 2014*). In regard to its conservation status, it has been listed under different conservation categories according to a variety of regional and global red

lists (*Bilz et al., 2011*; *Domina, 2011*; *Rossi et al., 2013*; *Urbani, Calvia & Pisanu, 2013*). Both species are conspicuous, tussock-forming plants, which are ecologically relevant as they are dominant and characteristic taxa in the plant communities where they live (phytosociological ranks: association, alliance, class; *Molina, 1996*; *Navarro, Molina & Moreno, 2001*; *Rufo Nieto & De la Fuente García, 2011*; *Gianguzzi et al., 2013*; *Rodríguez-Guitián et al., 2017*). Previous studies revealed a strong genetic structure in both species driven by geography, with distinct genetic clusters corresponding to disjunct populations (*Jiménez-Mejías et al., 2011*; *Benítez-Benítez et al., 2018*).

In the present study, we analyse the genetic makeup and potential distribution in these two ecologically relevant *Carex* sister species endemic to the Western Mediterranean Basin. We examine the relationship between these factors in a comparative and geographic-temporal framework in order to forecast the effects of GCC on these species. Thus, the main aims of this study are to: (1) estimate and compare the genetic diversity/ structure and potential distribution within and between the species in the present and future; (2) tackle the hypothesis that pose a negative relationship between range size, genetic diversity and ecological niche breadth on one side, and vulnerability to environmental changes on the other; (3) relate the current and future projected situation with the conservation status of both species.

# MATERIALS AND METHODS

## Analyses of present population genetics

We reanalysed the AFLP sampling that included 182 polymorphic loci used by *Benítez-Benítez et al. (2018)*, which consisted in 130 individuals from 18 different locations (69 individuals from 12 populations of *C. reuteriana* and 61 individuals from six populations of *C. panormitana*; Table 1), that representatively cover the range of both species fairly well.

Present gene diversity (Nei's diversity; *Nei, 1978*) at taxa and population level was computed with AFLP$_{DAT}$ (*Ehrich, 2006*) as implemented in R v. 3.2.1 (*R Core Development Team, 2020*). Since many *Carex* species often propagate vegetatively, including the two study species (*Luceño & Jiménez-Mejías, 2008*; P. Jiménez-Mejías & M. Urbani, 2021, personal communications), we tried to minimize sampling ramets from the same clone (genet) by selecting individuals at a distance of at least 2 m. The distribution of the number of pairwise genetic distance comparisons among phenotypes and individuals within a species representing putative clones were inferred with the clone function also in AFLP$_{DAT}$. The Arlequin function from AFLP$_{DAT}$ was used to calculate gene diversity of populations at the ramet and genet levels before and after removing putative clones, respectively. These clones were identified as phenotypes that differed in a number of bands below the frequency of the error rate (*Bonin et al., 2004*). Frequency-down-weighted marker values (DW) were also estimated as a measure of allele rarity (*Schönswetter & Tribsch, 2005*).

To further investigate the genetic structure found by *Benítez-Benítez et al. (2018)*, a phylogenetic tree of AFLP phenotypes was obtained with a Neighbor-Joining (NJ) analysis based on Nei-Li distances including 1,000 replicates to assess bootstrap support, as

**Table 1 Geographic location and voucher of each sampled population of *C. reuteriana* (two subspecies) and *C. panormitana*, average gene diversity at ramet and genet level for present and future (2081–2100) under RCP2.6 GCC scenario, and allelic rarity levels (DW).**

| Taxon/ Population | Locality | Voucher/ Herbarium | Number of sampled individuals | Longitude/ Latitude | DW present (genet) | Mean gene diversity ± SD (present; ramet level) | Mean gene diversity ±SD (present; genet level) | Mean gene diversity ± SD (future; ramet level) | Mean gene diversity ±SD (future; genet level) |
|---|---|---|---|---|---|---|---|---|---|
| *C. reuteriana* | | | 69 | | 905.394 | 0.184 | | 0.176 | |
| *C. reuteriana* ssp. *reuteriana* | | | 29 | | 154.041 | 0.142 | | 0.107 | |
| REU_POR-TM_1 | Portugal, Tras os Montes, Lamego, Bigorne, Petrarouca | M. Escudero et al., 37ME07 (UPOS-7374) | 6 | −7.88/ 41.03 | 13.558 | 0.106 ± 0.064 | | | |
| REU_POR-BL_2 | Portugal, Beira Litoral, Coimbra, Lousã | M. Escudero et al., 60ME07 (UPOS-7373) | 7 | −8.23/ 40.10 | 11.873 | 0.083 ± 0.048 | | | |
| REU_SPA-Av_3 | Spain, Ávila, Sierra de Gredos, Las Chorreras del Tormes | J.M. Marín, 5504JMM (UPOS-1004) | 5 | −5.16/ 40.34 | 25.219 | 0.110 ± 0.069 | | IAE | |
| REU_SPA-CcN_4 | Spain, Cáceres, Valley of Jerte | P. Jiménez-Mejías & I. Pulgar, 57PJM07 (UPOS-6957) | 4 | −5.75/ 40.22 | 9.278 | 0.114 ± 0.077 | | IAE | |
| REU_SPA-CcS_5 | Spain, Cáceres, Ibor river | P. Jiménez-Mejías et al., 24PJM13 (UPOS-5449) | 4 | −5.44/ 39.62 | 21.162 | 0.183 ± 0.122 | | IAE | |
| REU_SPA-To_6 | Spain, Toledo, Navalucillos | P. Jiménez-Mejías et al., 60PJM13 (UPOS-5479) | 3 | −4.66/ 39.64 | 9.189 | 0.158 ± 0.120 | | IAE | |
| *C. reuteriana* ssp. *mauritanica* | | | 40 | | 324.723 | 0.170 | | 0.164 | |
| MAU_SPA-Se_7 | Spain, Sevilla, El Ronquillo, Rivera de Huelva | P. Jiménez-Mejías, 35PJM07 (UPOS-7372) | 7 | −6.17/ 37.67 | 21.125 | 0.155 ± 0.089 | | | |
| MAU_SPA-CaGu_8 | Spain, Cádiz, El Gastor, Guadalete river | P. Jiménez-Mejías, 34PJM07 (UPOS-7371) | 5 | −5.45/ 36.88 | 8.625 | 0.100 ± 0.063 | | | |

*(Continued)*

| Taxon/ Population | Locality | Voucher/ Herbarium | Number of sampled individuals | Longitude/ Latitude | DW present (genet) | Mean gene diversity ± SD (present; ramet level) | Mean gene diversity ±SD (present; genet level) | Mean gene diversity ± SD (future; ramet level) | Mean gene diversity ±SD (future; genet level) |
|---|---|---|---|---|---|---|---|---|---|
| MAU_SPA-CaAl_9 | Spain, Cádiz, Alcornocales Natural Park, | P. Jiménez-Mejías & I. Pulgar, 17PJM07 (UPOS) | 8 | −5.59/ 36.55 | 24.108 | 0.084 ± 0.048 | | | |
| MAU_SPA-J_10 | Spain, Jaén, Despeñaperros | P. Jiménez-Mejías & L. Reina, 67PJM09 (UPOS) | 8 | −3.06/ 38.39 | 25.558 | 0.103 ± 0.058 | | IAE | |
| MAU_MOR-Lao_11 | Morocco, Tanger, Rif, Oued Laou | A.J. Chaparro et al., 8AJC05 (UPOS-1637) | 5 | −5.30/ 35.14 | 13.032 | 0.131 ± 0.081 | | | |
| MAU_MOR-Lou_12 | Morocco, Tanger, Rif, Oued Loukos | A.J. Chaparro et al., 3AJC05 (UPOS-1630) | 7 | −5.44/ 35.03 | 24.417 | 0.114 ± 0.066 | | | |
| *C. panormitana* | | | 61 | | 436.028 | 0.128 | 0.133 | 0.118 | 0.100 |
| Tunisia-Sicily | | | 23 | | 114.057 | 0.115 | | | |
| PAN_TUN_13 | Tunisia, Jendouba, El Feija National Park | P. Jiménez-Mejías & J.E. Rodríguez, 132PJM13 (UPOS-6636) | 9 | 8.31/36.49 | 36.949 | 0.067 ± 0.038 | | IAE | |
| PAN_SIC_15 | Italy, Sicily, Fiume Oreto | D. Cusimano s. n. (SS) | 14 | 13.34/ 38.09 | 51.900 | 0.077 ± 0.041 | | | |
| Sardinia | | | 38 | | 110.234 | 0.032 | 0.037 | | |
| PAN_SAR-Bau_16 | Italy, Sardinia, Bau Mela river, Villagrande | M. Urbani, 2013 (SS) | 9 | 9.42/39.98 | 11.239 | 0.024 ± 0.015 | 0.033 ± 0.022 | 0.024 ± 0.015 | 0.033 ± 0.022 |
| PAN_SAR-Pira_17 | Italy, Sardinia, Cantoniera, Pirae´onni, Villagrande | M. Urbani, 2013 (SS) | 10 | 9.40/40.02 | 7.471 | 0.026 ± 0.016 | 0.031±0.021 | 0.026 ± 0.016 | 0.031 ± 0.021 |
| PAN_SAR-Ber_18 | Italy, Sardinia, Ramacaso river, Berchidda | M. Urbani, 2013 (SS) | 8 | 9.24/40.82 | 7.934 | 0.023 ± 0.015 | | | |
| PAN_SAR-Cal_19 | Italy, Sardinia, Miriacheddu river, Calangianus | M. Urbani, 2013 (SS) | 11 | 9.26/40.89 | 7.577 | 0.012 ± 0.008 | 0.028 ± 0.030 | 0.012 ± 0.008 | 0.028 ± 0.030 |

Note:
Labelling of the populations specifies the taxa (REU, *Carex reuteriana* subsp. *reuteriana;* MAU, *Carex reuteriana* subsp. *mauritanica;* PAN, *Carex panormitana*), and the TDWG botanical country abbreviation (*Brummit, 2001*) (MOR, Morocco; POR, Portugal; SAR, Sardinia; SIC, Sicily; SPA, Spain; TUN, Tunisia). Herbarium acronyms are according to Index Herbariorum (*Thiers, 2020*). Populations inferred as extinct (IAE) under RCP2.6 scenario for 2081–2100 are not included in the genetic diversity calculation.

implemented in PAUP v. 4.0b10 (*Swofford, 2002*). An analysis of molecular variance (AMOVA) was conducted using ARLEQUIN v. 3.5.2.2 (*Excoffier & Lischer, 2010*) in which we also calculated the fixation index $F_{st}$ (*Wright, 1965*). We conducted two partitions in order to examine in detail the variation of genetic diversity, one considering the two species (*C. reuteriana* and *C. panormitana*), and the other partition considering the four main genetic clusters previously found by *Benítez-Benítez et al. (2018)*: (1) *C. reuteriana* subsp. *reuteriana*, (2) *C. reuteriana* subsp. *mauritanica*, (3) Sicilian-Tunisian populations of *C. panormitana*, and (4) Sardinian populations of *C. panormitana*. We also used BAPS v. 6.0 (*Corander, Waldmann & Sillanpää, 2003*) to look for possible underlying fine structure, estimating the number of genetic clusters (K) by assigning individuals and populations in undefined mixture clusters under a Bayesian framework. We ran the analysis with a predefined 10 replicates from K = 5 to K = 8. Finally, Discriminant Analysis of Principal Components (DAPC; *Jombart, 2008*) is a multivariate method that, unlike Principal Component Analysis (PCA) and Discriminant Analysis (DA), maximizes the separation between groups while minimizing variation within groups to identify genetic clusters (*Jombart, Devillard & Balloux, 2010*). It was used to obtain a visual spatial assessment of genetic structure using the four main genetic clusters previously defined as priors. In view of the weak genetic structure found within *C. reuteriana* subsp. *reuteriana* (see results), a Mantel test was performed with GenAlEx v. 6.5 (*Peakall & Smouse, 2012*) to evaluate the correlation between genetic (*Nei, 1972*) and geographic distances between its populations.

In order to enable the comparison of present genetic structure with that inferred for the future for the taxonomically-driven partition considering three groups (*C. reuteriana* subsp. *reuteriana*, *C. reuteriana* subsp. *mauritanica*, and *C. panormitana*), we conducted an analysis with Structure v. 2.3.4 (*Pritchard, Stephens & Donnelly, 2000*). For each taxon, we performed ten independent runs of 100,000 iterations each one, with a burn-in period of 1,000 for each value of K from 1 to 3. The best K was chosen comparing the probabilities for the K values inferred with Structure Harvester (*Earl & vonHoldt, 2012*) and the graphic was performed with Structure Plot v. 2.0 (*Ramasamy et al., 2014*).

## Potential species distribution and ecological niche modeling for present times

We used the same occurrence dataset (316 georeferenced records of *C. reuteriana* subsp. *reuteriana*, 118 of subsp. *mauritanica*, and 29 (all known records) of *C. panormitana*; Table S1) and uncorrelated bioclimatic variables used in *Benítez-Benítez et al. (2018)*: bio2 (mean diurnal range), bio4 (temperature seasonality), bio15 (precipitation seasonality) and bio16 (precipitation of wettest quarter). Distribution models were obtained for the three studied taxa: *C. reuteriana* subsp. *reuteriana*, *C. reuteriana* subsp. *mauritanica*, and *C. panormitana*. We used three groups (taxonomically-driven partition) rather than four (see genetic analyses) because of the low number of occurrences for *C. panormitana*.

Biomod2 (*Thuiller et al., 2009*) implemented in R was used for SDM, testing six different implemented modeling algorithms: Classification Tree Analysis (CTA), Generalized Additive Model (GAM), Generalized Boosted Regression Model (GBM), Generalized

Linear Model (GLM), Maximum Entropy Algorithm (MaxEnt), and Random Forest (RF). Thereafter, an ensemble modeling was run including all algorithms in order to build more accurate projections (*Araújo & New, 2007*; *Forester, DeChaine & Bunn, 2013*).

We randomly built three sets of pseudo-absences and generated a data splitting (80% training data and 20% burn-in) to assess the models by cross-validation, with two independent runs. We used True Skill Statistics (TSS; *Allouche, Tsoar & Kadmon, 2006*) and Area Under the Curve (AUC; *Swets, 1988*) with a threshold >0.7 as evaluation metrics for building models.

Finally, we conducted a Principal Component Analysis (PCA; *Janžekovič & Novak, 2012*) of the retained bioclimatic variables using the prcomp function in the package ggplot2 in R (*Wickham, 2016*) for visualizing the ecological niche occupied by each taxon (*C. reuteriana* subsp. *reuteriana*, *C. reuteriana* subsp. *mauritanica*, and *C. panormitana*) under present conditions.

## Future projections of potential distribution, genetic diversity/structure, and environmental niche

Future potential distribution was modeled with the same methodology as for the present (see above). Future climate can be projected with different general circulation models (GCMs) which are the primary source for studying the dynamics and components of the global climate system. GCMs were selected following *Parding et al. (2020)*: BCC-CMS2-MR, CanESM5, CNRM-CM6-1, CNRM-ESM2-1, MIROC6 and MRI-ESM2-0 (data available in https://www.worldclim.org/; *Fick & Hijmans, 2017*). These GCMs include four different Representative Concentration Pathways (RCPs) addressed in the IPCC Fifth Assessment Report (*IPCC, 2014*). We selected RCP2.6 and RCP8.5 to represent two extreme scenarios (an optimistic with lowest greenhouse gas atmospheric concentration *vs* a pessimistic with highest concentration). In addition, two future temporal ranges (2041–2060, 2081–2100) were used to project the set of GCMs. We have averaged the six different GCMs per each of the two scenarios resulting from modeling analyses in a single raster layer for each of the two future temporal ranges. This assembly accounts for uncertainty of future climatic conditions across the potential distribution range of species, as shown by *Wróblewska & Mirski (2018)*. We overlapped present and future projections of potential distribution in a single map using QGIS v. 3.4.15 (*QGIS Development Team, 2021*), enabling the visualization of the potential loss and/or gain of climatic suitable areas under different GCC scenarios. Thus, SDM have been used to show how the different Mediterranean regions where *C. reuteriana* and *C. panormitana* inhabit are responding to the GCC comparing current distribution models against the future ones in order to visualize the proportion of area loss/gain. It is important to note that this methodological approach assumes no migration and that species ecological requirement would remain the same (niche conservatism) in the future.

To estimate future gene diversity, we recalculated the values after removing those sampled populations placed within the predicted lost range according to the inferred potential distribution under the optimistic (RCP2.6) climate change scenario for 2081–2100. Therefore, we assumed the extinction of populations located in unsuitable

areas by future projections of SDM (*Espíndola et al., 2012*; *Lima et al., 2017*; *Rizvanovic et al., 2019*). We did not obtain genetic diversity values for the pessimistic (RCP8.5) 2081–2100 scenario, since all sampled populations of *C. panormitana* and *C. reuteriana* subsp. *reuteriana* were predicted to go extinct (see 'Results').

We used the spatial Bayesian clustering algorithm implemented in POPS v. 1.2 (*Jay, 2012*; *Jay et al., 2015*) to infer population genetic structure in the future (2081–2100) also based on the optimistic climate change scenario. We performed analyses for each taxon (*C. reuteriana* subsp. *reuteriana*, *C. reuteriana* subsp. *mauritanica*, and *C. panormitana*) with and without admixture. Two independent replicates were run for the best number of clusters (K = 2) obtained in Structure analyses for present times (see above), using genetic data from AFLPs, as well as spatial coordinates of each population and quantitative bioclimatic variables as covariates. We selected the same bioclimatic variables utilized in the future projections of SDM described above. For the simulated data, we used 10,000 sweeps with a burn-in of 2,000 sweeps to run POPS. The resulting output files were introduced in CLUMPP v. 1.1.2 (*Jakobsson & Rosenberg, 2007*) in order to average two runs per each K. Subsequently, the outputs from CLUMPP were imported in Distruct v. 1.1 (*Rosenberg, 2004*) to visualize the genetic structure for each independent genetic cluster analysed.

Finally, we projected the future environmental space of each taxon. A PCA was performed in the same way as for contemporary conditions, but removing those occurrences located in areas of lost habitat as explained for the estimation of future genetic diversity (optimistic scenario in 2081–2100).

## RESULTS

### Present genetic structure and diversity

NJ analysis of AFLP phenotypes recovered three moderately to well-supported clades (Fig. 1A): (1) *C. reuteriana* (88% Bootstrap (BS)), (2) Sicilian-Tunisian *C. panormitana* populations (75% BS), and (3) Sardinian *C. panormitana* populations (100% BS). In turn, Sicilian and Tunisian populations were recovered as sisters in well-supported distinct clades (98% BS Tunisia, 92% BS Sicily). On the contrary, subspecies and populations of *C. reuteriana* were unresolved in a polytomy.

AMOVA analysis results are shown in Table 2. The complete dataset considering the two species revealed that 28.06% of the variation of genetic diversity is explained by differences between the species, 37.22% by differences among populations within species, and 34.73% by differences within populations. Considering each species separately, variation among populations in *C. reuteriana* was considerably lower (39.82%) than in *C. panormitana* (68.76%), so variation within populations showed the opposite pattern (60.18% *vs* 31.24%, respectively). When partitioning the four main genetic clusters previously found by *Benítez-Benítez et al. (2018)*, AMOVA analysis revealed 42.13% of the variation of genetic diversity is explained by differences between groups, 21.46% by differences among populations within groups, and 36.42% by differences within populations. Regarding *C. reuteriana*, the differentiation between populations was higher in subspecies *mauritanica* (37.02%) than in subspecies *reuteriana* (20.36%).

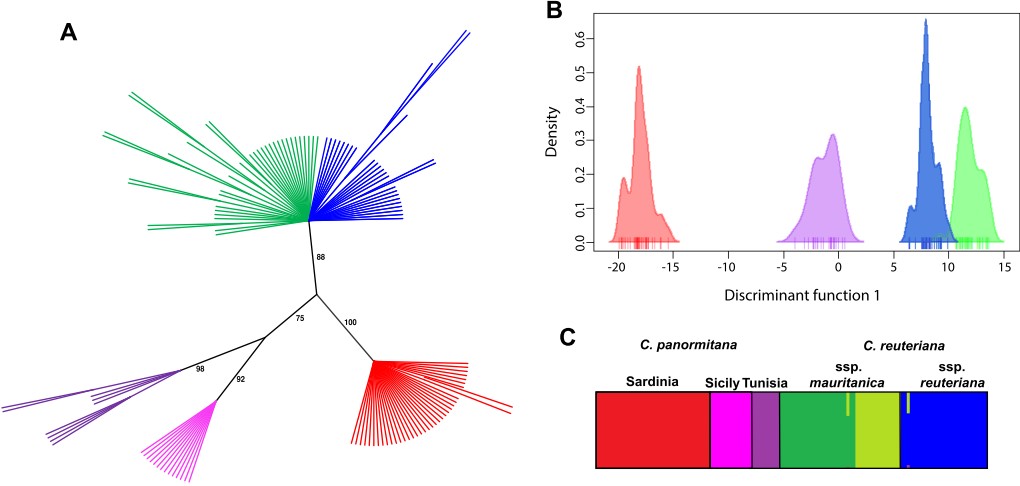

**Figure 1 Main results of the analyses of present genetic structure in *C. reuteriana-C. panormitana*.**
(A) Neighbor-joining tree based on Nei-Li genetic distances obtained from AFLP phenotypes for all sampled individuals; numbers above branches indicate bootstrap values (>50%); (B) DAPC scatter plot showing the discriminant function (axis x) and density (axis y) across two principal components; (C) BAPS admixture bars obtained for all sampled individuals, each represented by a bar. Colours represent the taxonomic or geographic adscription of samples: *C. reuteriana* subsp. *reuteriana* (blue), *C. reuteriana* subsp. *mauritanica* (green), and *C. panormitana* (red -Sardinia-, pink -Sicily-, and purple -Tunisia-). In DAPC (B), purple represents both Sicilian and Tunisian populations of *C. panormitana*; and lines below each density plot represent individuals. In BAPS (C), two shades of green are used to represent underlying geographic structure found within *C. reuteriana* subsp. *mauritanica* (dark green -northern Africa and south of the Guadalquivir Valley populations-, light green-Sierra Morena populations-).

In *C. panormitana*, there was a stronger difference between Sicilian-Tunisian populations (53.51%) than among Sardinian ones (41.4%).

DAPC analysis also confirmed the clear differentiation between the four genetic clusters previously commented (Fig. 1B), and a perfect assignment of each individual to its *a priori* group. However, BAPS analyses found a certain degree of additional, underlying genetic structure in *C. reuteriana* subsp. *mauritanica* and *C. panormitana*, and suggested that six is the most accurate number of genetic clusters (K = 6; Fig. 1C). Three of these clusters corresponded to *C. reuteriana*: (1) all six *C. reuteriana* subsp. *reuteriana* sampled populations; (2) four populations of *C. reuteriana* subsp. *mauritanica*, two located in northern Africa (Morocco) and the other two from the Betic ranges south of Guadalquivir Valley in southern Spain; and (3) the remaining two subsp. *mauritanica* populations from Sierra Morena (north of Guadalquivir Valley). *Carex panormitana* was also split into three clusters: (4) the four populations from Sardinia, (5) the population from Sicily and (6) the population from Tunisia.

STRUCTURE retrieved two as the optimal number of genetic clusters for the three taxa (K = 2; Fig. 2A). *Carex reuteriana* subsp. *reuteriana* clusters lacked geographic structure and presented admixture in all sampled populations. The Mantel test was not significant for this subspecies (*p*-value = 0.46; R2 = 0.001; Fig. S2 in Supplemental Material). On the contrary, *C. reuteriana* subsp. *mauritanica* presented a clearer genetic

**Table 2 AMOVA analyses for AFLPs data.**

| Grouping compared and source of variation | d.f. | Sum of squares | Variance components | Percentage of variation |
|---|---|---|---|---|
| 1-Whole dataset (two groups: *C. reuteriana* s.l. *vs C. panormitana*) | | | | |
| Among groups | 2 | 622.957 | 6.24105 | 28.06% |
| Among pops. | 15 | 800.836 | 6.46433 | 37.22% |
| Within pops. | 112 | 798.007 | 7.12506 | 34.73% |
| *C. reuteriana* | | | | |
| Among pops. | 10 | 369.104 | 4.65780 | 39.82% |
| Within pops. | 57 | 593.411 | 10.41071 | 60.18% |
| *C. panormitana* | | | | |
| Among pops. | 3 | 44.146 | 1.16277 | 68.76% |
| Within pops. | 55 | 204.596 | 3.71993 | 31.24% |
| 2-Whole dataset (four groups: *C. reuteriana* ssp. *reuteriana vs C. reuteriana* spp. *mauritanica vs C. panormitana* (Sicily + Tunisia) *vs C. panormitana* (Sardinia) | | | | |
| Among groups | 3 | 920.170 | 8.24129 | 42.13% |
| Among pops. | 14 | 503.623 | 4.19747 | 21.46% |
| Within pops. | 112 | 798.007 | 7.12506 | 36.42% |
| *C. reuteriana* ssp. *reuteriana* | | | | |
| Among pops. | 5 | 117.297 | 2.70605 | 20.36% |
| Within pops. | 23 | 243.393 | 10.58230 | 79.64% |
| *C. reuteriana* ssp. *mauritanica* | | | | |
| Among pops. | 5 | 251.807 | 6.05238 | 37.02% |
| Within pops. | 34 | 350.018 | 10.29464 | 62.98% |
| *C. panormitana* (Sicilian-Tunisian populations) | | | | |
| Among pops. | 1 | 90.374 | 7.64231 | 53.51% |
| Within pops. | 21 | 139.452 | 6.64059 | 46.49% |
| *C. panormitana* (Sardinian populations) | | | | |
| Among pops. | 3 | 44.146 | 1.35354 | 41.40% |
| Within pops. | 34 | 65.144 | 1.91599 | 58.60% |

**Note:**
The first group includes for comparisons the *Carex reuteriana-C. panormitana* complex and for each species separately. The second group compares the four AFLPs groups found.

structure, with the two clusters mostly corresponding to Iberian populations from north of the Guadalquivir Valley, and south of the valley (Betic ranges) plus North Africa, respectively. Finally, the two genetic clusters in *C. panormitana* corresponded, almost without admixture, to Sardinian populations and Sicilian-Tunisian ones, respectively.

The distribution of the number of pairwise differences among AFLP individual phenotypes in both species was unimodal for *C. reuteriana* while bimodal for *C. panormitana* (Fig. S1). The latter indicated the existence of clones in this species. In congruence, three of the four sampled Sardinian *C. panormitana* populations showed a significant presence of putative ramets (18 of 38 sampled individuals), according to the clone function in AFLP$_{DAT}$. The final AFLP error rate was 1.44% when clones were removed.

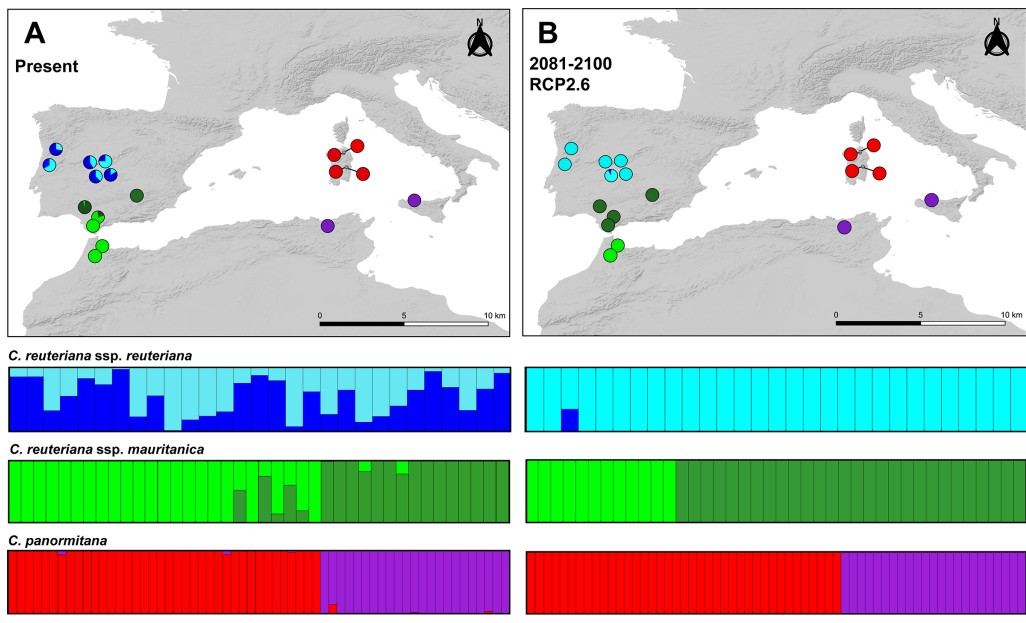

**Figure 2 Population genetic structure analyses.** (A) Population genetic structure for present times inferred by STRUCTURE, and (B) for future conditions (2081–2100) under RCP2.6 scenario projected with POPS. Circles indicate sampled populations of *C. reuteriana* subsp. *reuteriana*, *C. reuteriana* subsp. *mauritanica*, and *C. panormitana*. Colours are as in Fig. 1 and represent the identified genetic groups (K = 2 for the three taxa). Pie charts show the proportion of individuals in each population assigned to each of the two genetic groups. Likewise, colour bars represent the adscription of all sampled individuals to each of the two genetic groups detected for each taxa.

Finally, the analysis of the 130 sampled individuals from 18 populations revealed a high genetic differentiation between *C. reuteriana* and *C. panormitana* with an $F_{st}$ value of 0.653. Values were also high when considering *C. reuteriana* subspecies ($F_{st}$ = 0.46) and Sicilian-Tunisian *vs* Sardinian populations of *C. panormitana* ($F_{st}$ = 0.765).

Values of present genetic (Nei's) diversity for each taxon and sampled population are shown in Table 1. *Carex reuteriana* s.l. (0.184), as well as both its subspecies (subsp. *reuteriana*: 0.142; subsp. *mauritanica*: 0.170), have higher genetic diversity than *C. panormitana*. For the latter, the presence of putative clones was reflected in slightly different values at the genet (0.133) and ramet (0.128) levels. These clones were only detected for Sardinian populations, which was reflected in a very low genetic diversity in these populations (0.037 at genet and 0.032 at ramet level) in comparison with Sicilian-Tunisian counterparts (0.115). *Carex panormitana* as a whole presented higher DW value (436.028) in comparison to both subspecies of *C. reuteriana* (subsp. *reuteriana*: 154.041; subsp. *mauritanica*: 324.723), whilst *C. reuteriana* s.l. retrieved the highest DW value (905.394). In addition, *C. reuteriana* subsp. *reuteriana* displayed lower DW values than subsp. *mauritanica* in most of its populations (Table 1).

## Future genetic structure and diversity

The POPS analyses projecting the future genetic structure did not detect any new genetic clusters with respect to STRUCTURE analyses for present times (Fig. 2B), either

implementing admixture or non-admixture analyses. However, it revealed substantial changes in the genetic structure of both subspecies of *C. reuteriana*. Thus, *C. reuteriana* subsp. *reuteriana* was projected to almost completely lose one of its two current genetic clusters (except for one individual). In *C. reuteriana* subsp. *mauritanica*, all Iberian sampled populations were assigned to the same genetic cluster, while the other cluster would remain exclusively for Moroccan populations, in both cases with no admixture between clusters. In contrast, *C. panormitana* was projected to retain the same strong genetic structure found for present times, which was even reinforced by the loss of the marginal admixture.

Future projections of genetic diversity removing populations affected by habitat loss according to SDM results (see below; RCP2.6 for 2081–2100 in Fig. 3) yielded lower values of genetic diversity when compared to present for all taxa, although only slightly for *C. reuteriana* s.l. (*C. reuteriana* subsp. *reuteriana*: 0.107; *C. reuteriana* subsp. *mauritanica*: 0.164; *C. panormitana*: 0.118 (genet), 0.100 (ramet)). DW values were considerably higher in the Sicilian, and, to a lesser extent, in the Tunisian population of *C. panormitana* than in the Sardinian ones and both subspecies from *C. reuteriana* (Table 1).

## Species distribution modeling

Projected potential distribution ranges of *C. reuteriana* and *C. panormitana* shifted according to six GCMs in different time periods (2041–2060 and 2081–2100) and two RCPs scenarios (RCP2.6 and RCP8.5). Both species were inferred to gain and lose suitable areas in response to GCC under different scenarios (see projections for 2041–2060 in Fig. S3 in Supplemental Material and Fig. 3 for 2081–2100). Herein, we will mainly comment on the results from 2081–2100 since this scenario represents more severe climatic conditions in comparison with current ones (Fig. 3).

Future SDM revealed important losses of suitable areas for all taxa, which was reinforced in the RCP8.5 scenario, especially for *C. panormitana*. Likewise, currently suitable areas inferred to remain stable (current potential range overlapping with that inferred by future models) were always reduced in the RCP8.5 with respect to RCP2.6. In *C. reuteriana* subsp. *reuteriana* (Figs. 3A, 3B) climate change was predicted to produce the loss of around the half of its currently suitable areas (41% in the RCP2.6 scenario and 57% in the RCP8.5 scenario), including all the Central Iberian range, and, in the RCP8.5, also the NW Iberian quadrant. The most stable areas (40% RCP2.6, 10% RCP8.5) would be located in the NW Iberian Peninsula. Future suitable conditions (22% RCP2.6, 33% RCP8.5) could spread northwards of its current distribution (Atlantic coast of North Spain and Western France; Fig. 3A). *Carex reuteriana* subsp. *mauritanica* (Figs. 3C, 3D) displayed a predicted loss of potential area in its current range in Sierra Morena (34% RCP2.6, 66% RCP8.5), while stable areas (53% RCP2.6, 24% RCP8.5) could persist south of Guadalquivir Valley and in the Tingitan Peninsula. Future suitable areas also extended to the north Atlantic coast of Portugal for both scenarios (13% RCP2.6, 10% RCP8.5). *Carex panormitana* (Figs. 3E, 3F) yielded a remarkable reduction of its potential distribution area under the two future climate scenarios (66% RCP2.6, 95% RCP8.5) with respect to its current distribution range. The potentially future suitable areas were very low (8% RCP2.6,

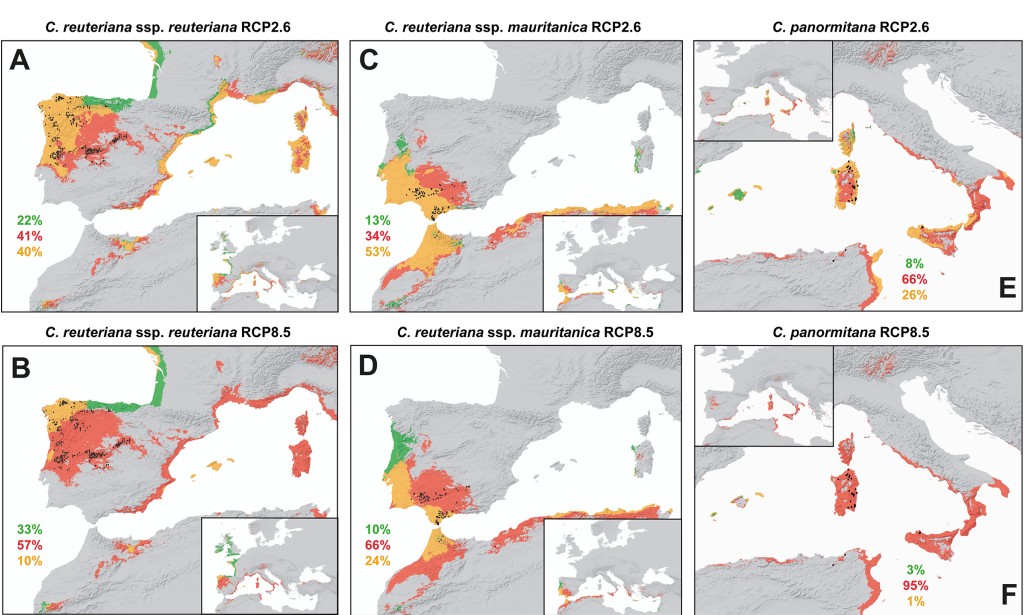

**2081-2100**

**Figure 3 Predicted changes of potential distribution inferred by Biomod.** Projections comparing present and future times (2081–2100) under RCP2.6 (top maps) and RCP8.5 (bottom maps) climatic change scenarios for (A, B) *C. reuteriana* subsp. *reuteriana*, (C, D) *C. reuteriana* subsp. *mauritanica*, and (E, F) *C. panormitana*. Future projections represent the consensus SDM averaged across six GCMs. Percentages indicate the proportion of potential range inferred to be gained, lost, or to remain stable in the future in comparison with the present, according to the following colour scheme: red areas are currently suitable areas predicted to be lost, orange areas are currently suitable areas projected to remain stable, and green areas are currently unsuitable areas projected to become suitable. Black dots represent current occurrences of each taxon used for SDM.

3% RCP8.5), mostly recovered in Balearic Islands (Fig. 3E, Fig. S3). A 26% of stable areas were found in the coast of Tyrrhenian Islands (Corsica, Sardinia, and Sicily) for the RCP2.6, whilst the RCP8.5 scenario only showed 1%.

PCAs showed (see Fig. S4) a similar environmental space both for present and future in 2081–2100 under RCP2.6 scenario, with also analogous percentages for the explained variances along their PC1s (54.2% and 56.2%, respectively) and PC2s (27.8% and 23.5%, respectively). *Carex reuteriana*, including its subspecies, displayed a wider environmental niche than *C. panormitana* both in the present and the future.

# DISCUSSION

## Geographically-driven strong genetic structure and contrasting patterns of genetic diversity

The strong genetic structure for the present time (Figs. 1, 2A) supports the genetic clusters previously obtained by *Jiménez-Mejías et al. (2011)* and *Benítez-Benítez et al. (2018)*. AFLPs revealed a slight differentiation between the two subspecies of *Carex reuteriana*, in congruence with their allopatric distribution (Fig. 1), suggesting the importance of geographic speciation during their differentiation (*Cabej, 2012*; *Sobel, 2016*). For the subspecies *reuteriana*, DAPC and BAPS analyses recovered a single genetic cluster likely

due to gene flow between its populations, whilst for subspecies *mauritanica* two genetic clusters with geographical congruence were identified (Figs. 1, 2A). Thus, the latter subspecies seems to have been influenced by the role of Guadalquivir Valley as a barrier to dispersal since there exists a clear genetic differentiation between Sierra Morena and Betic populations (Figs. 1C, 2A, Table 2). This pattern was previously pointed out in other studies about the sect. *Phacocystis* (*Jiménez-Mejías et al., 2011*; *Benítez-Benítez et al., 2018*), as well as in other Mediterranean plant groups (*Ortiz et al., 2008*; *Casimiro-Soriguer et al., 2010*; *Tremetsberger et al., 2016*; *Fernández i Marti et al., 2018*). By contrast, populations at both sides of the Strait of Gibraltar display a higher genetic similarity (Figs. 1C, 2A), supporting this area as a single connected refuge (*Marañón et al., 1999*; *Arroyo et al., 2008*; *Rodríguez-Sánchez et al., 2008*; *Molina-Venegas, 2015*). The higher DW value and genetic diversity in the subspecies *mauritanica* (324.723, 0.170; see Table 1) than subspecies *reuteriana* (154.041, 0.142; Table 1) seems to correspond with the higher genetic structure found across the populations of the first subspecies. This higher allelic rarity identified in the subspecies *mauritanica* from these southernmost, rear-edge populations of the Iberian Peninsula could be a reflect of their long-term isolation and persistence in refugia during Pleistocene glaciations (*Médail & Diadema, 2009*; *Provan & Maggs, 2011*; see Last Glacial Maximum projections in *Benítez-Benítez et al., 2018*).

On the other hand, *C. panormitana* displayed a significant genetic structure between Sicilian-Tunisian and Sardinian populations for all genetic analyses (Figs. 1, 2A), as shown in other Tyrrhenian endemics (*e.g.*, *Bittkau & Comes, 2005*; *Molins et al., 2018*). The genetic similarity found across the Strait of Sicily in different group of plants (*e.g.*, *Fernández-Mazuecos & Vargas, 2011*; *Lo Presti & Oberprieler, 2011*; *De Castro et al., 2015*; *Tremetsberger et al., 2016*), also identified in *C. panormitana* (Fig. 1), suggests an important dispersal route between Sicily and North of Africa. The genetic diversity in the Tyrrhenian *C. panormitana* is much lower than in its mainland sister *C. reuteriana* (0.133 and 0.184 respectively, Table 1), supporting the finding that species with wider distribution ranges combined with larger population sizes frequently display higher levels of genetic diversity than restricted endemic plants (*López-Pujol et al., 2009*, *2013*; *García-Verdugo et al., 2015*; *Fernández-Mazuecos et al., 2016*). Nonetheless, high genetic diversity has been reported in some endemic species with narrow distributions from the Mediterranean region (*e.g.*, *Mameli et al., 2008*; *Mayol et al., 2012*; *Jiménez-Mejías et al., 2015*; *Fernández-Mazuecos et al., 2016*). Specifically, *C. panormitana* displays striking contrasting patterns between its disjunct populations, with lower genetic diversity but larger distribution range in Sardinia, while higher diversity and distinctiveness but extremely reduced distribution in Sicily and Tunisia (only one population each; *Urbani, Gianguzzi & Ilardi, 1995*; *Urbani, Calvia & Pisanu, 2013*). These higher DW values could be due to a long-term persistence and isolation of Sicilian-Tunisian populations. This agrees with Sicily and NE Tunisia serving as glacial refugia during the Last Glacial Maximum (*e.g.*, *Schönswetter et al., 2003*; *Magri et al., 2006*; *Médail & Diadema, 2009*; *Jiménez-Mejías et al., 2012*; see Last Glacial Maximum projections in *Benítez-Benítez et al., 2018*). Sardinian populations were the only that displayed a significant number of clones. Therefore, the clonality detected within them could be probably responsible at least in part

for its low genetic diversity. This low genetic diversity in Sardinian populations could have been also caused by a population bottleneck (see below).

## Future reduction of potential distribution and genetic diversity under GCC

Our climatic predictions show that at the end of the century the loss of distribution range will be clearly higher than the range gain for both Mediterranean species (Fig. 3; *e.g.*, *Casazza et al., 2014*; *Al-Qaddi et al., 2017*; *Vessella et al., 2017*; *Kougioumoutzis et al., 2020*). This could convey ecological consequences because *C. reuteriana* and *C. panormitana* play important roles within the ecosystem functioning of rivershores where they inhabit (*Rodríguez-Guitián et al., 2017*). Although the more restricted *C. panormitana* seems to be more negatively affected by climate change effects (Figs. 3E, 3F) than the more widespread *C. reuteriana* (Figs. 3A–3D), their potential distribution range will respond in similar ways under future GCC in terms of habitat loss. Given the ecological dominance of these two species in their respectives ranges, their disappearance could even alter the communities and boundaries of habitats that these species help define.

*Carex reuteriana* would maintain a larger distribution area (Fig. 3) and environmental space (Fig. S4), as well as greater genetic diversity (Table 1) than *C. panormitana*, despite many of its populations could be potentially wiped out in response to the GCC (Figs. 3A–3D; Fig. S3). Whether these populations would disappear, the dominant phytosociological associations (*Carici reuterianae-Betulum celtiberica*, *Rodríguez-Guitián et al., 2017* and *Caricetum tartessianae*, *Molina, 1996*; *Navarro, Molina & Moreno, 2001*; *Rufo Nieto & De la Fuente García, 2011*) that forms part of the Iberian riparian forests will be affected. *Carex reuteriana* subsp. *reuteriana* might be able to migrate and change its distribution range (*Bussotti et al., 2015*) whilst conserve its ecological niche according to the displacement of climatic suitability to northwards of its current distribution range (*e.g.*, *Alsos et al., 2012*; *Wróblewska & Mirski, 2018*). The relatively high levels of genetic diversity and ecological width of species may also enable *in situ* local adaptations, triggering high resilience to environmental changes and in turn buffering the loss of its genetic diversity (*Aubin et al., 2016*; *Bussotti & Pollastrini, 2017*; *Lima et al., 2017*). However, it has been reported that species might not be able to shift their distribution range toward suitable conditions as fast as the GCC is taking place (*Loarie et al., 2009*) and thus their survival could also depend on other factors (*e.g.*, phenotypic plasticity, adaptive capacity, dispersal or colonization ability; *Hoffmann & Sgrò, 2011*; *Razgour et al., 2019*). A significant loss of suitable potential areas is predicted for both subspecies of *C. reuteriana*, although with slight differences between them (Figs. 3A–3D), whilst their genetic structure also could suffer important changes in the future (2081–2100; Fig. 2). Specifically, the subspecies *mauritanica* could undergo gene flow limitation across the Strait of Gibraltar, strengthening genetic structure between Morocco and southernmost Iberian Peninsula populations (Fig. 2B; *e.g.*, *Escudero et al., 2008*; *Ortiz et al., 2008*; *Terrab et al., 2008*). In contrast, populations at both sides of the Guadalquivir Valley could increase their genetic admixture.

Otherwise, the more restricted *C. panormitana* seems to recover very little available potential habitat and great habitat loss as well as reduction in its genetic diversity in the future (Fig. 3; Table 1; Fig. S3). On the contrary, the current genetic structure of this Tyrrhenian endemism (Sicilian-Tunisian *vs* Sardinian populations, Fig. 2) could remain stable over time. Predictions are especially alarming under the pessimist emission scenario (95% RCP8.5, Fig. 3F), which indicate that this species could almost disappear in the future as a direct consequence of the GCC. The extinction of *C. panormitana* will probably affect the functioning of the ecosystem, as it dominates the riparian communities with *Carex pendula*, characterizing a phytosociological association (*Caricetum pendulo-panormitanae*, *Gianguzzi et al., 2013*). Our findings support the hypothesis that local extinction rates appear to be higher in species with restricted distribution, although it could also depend on their specific niche characteristics (*Lavergne, Molina & Debussche, 2006*; *Gray, 2018*). Specifically, the low genetic diversity found in *C. panormitana*, which would likely decrease in the future, foretells little capacity to adapt to the new climatic conditions and therefore to modify its niche, which may contribute to its extinction (*Dagnino et al., 2020*; *Olave et al., 2019*; Figs. 3E, 3F; Table 1; Fig. S4). However, a niche shift during the Pliocene has been proposed for this lineage of plants (*Benítez-Benítez et al., 2018*), which suggests that it may retain certain adaptation capacity when facing future climatic changes.

Although distribution models do not incorporate factors such as biotic interaction, dispersal or adaptation into future projections, our results demonstrate the usefulness of combining SDM and molecular genetic analysis to approach the future of species. Specifically, the Mediterranean Basin is considered a vulnerable hotspot, where plant extinction is already taking place (*Médail, 2017*; *Orsenigo et al., 2018*; *IUCN, 2021*) and as much as 3,000 species (*Malcolm et al., 2006*) have been predicted to become extinct in the future under GCC scenarios (*Bellard et al., 2014*). Likewise, the threat by desertification and expansion of arid regions (*IPCC, 2014*) are also being induced by global warming. Those habitats which have been inferred to remain mostly stable under ongoing GCC (*e.g.*, parts of southernmost areas of Iberian Peninsula, northern Africa, and Tyrrhenian Islands; *Médail & Diadema, 2009*) have been considered as important refugia of current diversity (*Temunović et al., 2013*). Nevertheless, the permanence of species in these areas will depend not only on the future climatic conditions but also the species ability to resist other threats like anthropogenic fragmentation or habitat loss (*Thomas et al., 2004*; *Malcolm et al., 2006*; *Newbold et al., 2015*; *Hoffmann et al., 2018*).

## Conservation and survival issues in *C. panormitana*

Our results express great concern about the conservation perspectives and future viability of *Carex panormitana*. A review of different assessments of its conservation status clearly points to a conflicting situation. Assessments at different geographical scales frequently result in the application of different criteria and conservation categories due to considering different parts of the taxon range. However, in the case of the restricted endemic *C. panormitana*, the whole range was considered. On the one hand, it has been listed as a species with conservation priority for the European Union in the Annex II of the Habitat

Directive 92/43/CEE, and it has also been catalogued as threatened in different regional red lists from Italy ("critically endangered (CR)" by *Conti, Manzi & Pedrotti (1997)*, and "endangered (EN)" by *Rossi et al. (2013)*, *Urbani, Calvia & Pisanu (2013)*). On the other hand, it has been classified as "least concern (LC)" in the European Red List of Vascular Plants (*Bilz et al., 2011*) and at the global level in the IUCN Red List (*Domina, 2011*). Importantly, the IUCN global assessment is based on a wrong distribution of the species (*Domina, 2011*), which includes regions of Greece as part of the distribution range of *C. panormitana*, whereas it is absent there (*Jiménez-Mejías et al., 2014*). This was probably due to a misleading identification with the closely related *C. acuta* L. or *C. kurdica* Kük. ex Hand.-Mazz., which is present in the Balkan Peninsula (*Jiménez-Mejías et al., 2014*). Thus, *C. panormitana* requires, in view of our results, a reassessment of its conservation status at the global level applying the IUCN criteria and guidelines (*IUCN, 2012*; *IUCN Standards & Petitions Committee, 2019*; respectively).

The only known Sicilian population is located in the outskirts of Palermo city (*Urbani, Gianguzzi & Ilardi, 1995*, *Urbani, Calvia & Pisanu, 2013*) along the shores of the Oreto River (*Gianguzzi et al., 2013*). This population displays high genetic distinctiveness (see DW values; Table 1) and has important historical value, as it is the type locality of the species (*Gussone, 1844*; see *Jiménez-Mejías et al., 2014*), but its habitat is considered especially sensitive to disturbances (*Thiébaut, 2006*). The proximity of this population to Palermo increases human pressure through different threats (*e.g.*, urban expansion, pollution of river water by agricultural and domestic activities, invasive species, etc.) leading to habitat degradation and fragmentation (*Gianguzzi et al., 2013*). Moreover, a fragmentation of this population and a decline in its size have been detected in recent years (*Gianguzzi et al., 2013*). Habitat fragmentation and small population size could produce loss of genetic diversity (Table 2) through processes like interruption of gene flow between subpopulations, inbreeding and genetic drift. The latter process could also entail the loss of rare alleles.

In view of our results, conservation measures should be urgently implemented to safeguard the future of *C. panormitana* and the habitats where it lives. The reinforcement of natural populations through the translocation of individuals between populations should be avoided due to the strong genetic differentiation and distinctiveness found between Sicilian-Tunisian and Sardinian populations. Accordingly, a specific conservation program should be designed for *C. panormitana* which should include collection of seeds (and propagules) and *ex-situ* storage in germplasm banks. Seed collecting (and *ex situ* growing of living individuals) should be conducted separately for each of the three regions where this species inhabits (Sardinia-Sicily-Tunisia), that should constitute different management units, to adequately represent its genetic diversity and maintain its genetic structure. *Carex panormitana* displays high levels of genetic diversity in Sicily and Tunisia, representing an important reservoir for conservation of genetic resources, whilst Sardinian populations show an important genetic impoverishment (Table 1). In some of the latter, the number of individuals could have severely decreased due to river floods

(M. Urbani, 2021, personal communications), perhaps producing bottlenecks that would help to explain the low genetic diversity. Likewise, there are also at least one population (Cantoniera Pirae´onni) in which overgrazing by cattle seems to be hindering flowering of individuals (M. Urbani, 2021, personal communications), which could reduce sexual reproduction and therefore perhaps affecting genetic diversity (*e.g.*, *Wu et al., 2010*; *Fernández-Mazuecos et al., 2016*; *Souto & Tadey, 2019*). Pursuant to the foregoing, it would be of paramount importance investing on projects for biodiversity conservation, maintenance of the structure and ecosystem functions, as well as decreasing the degree of disturbance in the habitat to guarantee its long-term survival.

On the other side, the distribution range of *C. panormitana* is poorly known in Tunisia, where its presence was not noted until a few years ago due to the confusion with other species of *Carex* sect. *Phacocystis* (*Jiménez-Mejías et al., 2014*). Prospections would be also desirable in NE Algeria, as one of the recently discovered Tunisian populations (*Jiménez-Mejías et al., 2014*) is fairly close to the border (see *Benítez-Benítez et al., 2018*). Therefore, further field and/or herbarium surveys are needed for this area of NW Africa to search for new populations which may influence its conservation status.

## CONCLUSIONS

Our results establish a clear genetic differentiation and strong structure both between and within *C. panormitana* and *C. reuteriana* (including both its subspecies *reuteriana* and *mauritanica*). The finding of genetic clusters according to disjunct areas suggests restricted gene flow among populations and a significant role of geographical barriers. *Carex reuteriana* showed higher levels of genetic diversity than *C. panormitana*, which presented the lowest values in Sardinia probably due to the importance of vegetative reproduction. On the contrary, *C. panormitana* displayed a greater genetic distinctiveness than both subspecies of *C. reuteriana*. Our future scenarios of climate change forecast a reduction in the genetic admixture and diversity in most populations of both species. In addition, SDM infer an overall loss of potential area for the three taxa, especially for *C. panormitana* which could lose almost its entire distribution range and even disappear under the most severe GCC scenario in the further future (RCP8.5 in 2081–2100). These results, combined with the conflicting conservation assessments previously proposed for *C. panormitana* and the extant threats to its persistence, support the urgent need to reassess globally the conservation status of this Tyrrhenian restricted endemic and implement *ex-situ/in-situ* conservation measures.

To sum up, this work displays how SDM in conjunction with molecular data can be used to forecast the effects of GCC on the potential distribution and future dynamics of genetic diversity and structure of species in the future. Therefore, this could be a useful approach for conservation management and planning, helping the allocation of resources for priority species and/or populations.

## ACKNOWLEDGEMENTS

The authors thank P. Vargas for his support during AFLPs procedures.

### Funding

This study was carried out with the financial support of the Spanish Ministry of Economy and Competitiveness (project CGL2016-77401-P) and the Spanish Ministry of Science and Innovation (project PID2020-113897GB-100) and the Fondo di Ateneo per la Ricerca 2019 (FAR 2019). Carmen Benítez-Benítez was supported by a Predoctoral Fellowship Program grant from the Ministry of Science, Innovation and Universities (FPU16/01257), and María Sanz-Arnal was funded by a Research Fellowship grant from Universidad Pablo de Olavide (PPI1903). The funders had no role in study design, data collection and analysis, decision to publish, or preparation of the manuscript.

### Grant Disclosures

The following grant information was disclosed by the authors:
Spanish Ministry of Economy and Competitiveness: CGL2016-77401-P.
Spanish Ministry of Science and Innovation: PID2020-113897GB-100.
Fondo di Ateneo per la Ricerca 2019.
Predoctoral Fellowship Program grant from the Ministry of Science, Innovation and Universities: FPU16/01257.
Research Fellowship grant from Universidad Pablo de Olavide: PPI1903.

### Competing Interests

Santiago Martín-Bravo is Academic Editor for PeerJ.

### Author Contributions

- Carmen Benítez-Benítez conceived and designed the experiments, performed the experiments, analyzed the data, prepared figures and/or tables, authored or reviewed drafts of the article, and approved the final draft.
- María Sanz-Arnal analyzed the data, prepared figures and/or tables, authored or reviewed drafts of the article, and approved the final draft.
- Malvina Urbani performed the experiments, authored or reviewed drafts of the article, participated in field trips, and approved the final draft.
- Pedro Jiménez-Mejías conceived and designed the experiments, performed the experiments, authored or reviewed drafts of the article, participated in field trips, and approved the final draft.
- Santiago Martín-Bravo conceived and designed the experiments, performed the experiments, authored or reviewed drafts of the article, participated in field trips, and approved the final draft.

### Data Availability

  The raw data are available in the Supplemental Files.

## Supplemental Information

Supplemental information for this article can be found online at http://dx.doi.org/10.7717/peerj.13464#supplemental-information.

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
