# Peer review of "Dramatic impact of future climate change on the genetic diversity and distribution of ecologically relevant Western Mediterranean Carex (Cyperaceae)"

_PeerJ, doi:10.7717/peerj.13464_

## Round 0.1 · original submission · Minor Revisions

Dear Authors

I received the evaluations of two reviewers, both very positive, and I also reviewed the manuscript and I agree with them. There are only a number of things that I think can be easily cleared up and incorporated into a new version of the manuscript. In addition to what has been highlighted by the reviewers, I ask you for a further effort as regards the part on conservation which, in the current version, is not adequately addressed; in particular:

- as pointed by reviewer 2, it's incorrect to state that its conservation status is disputed since assessments made on different geographical scales often produce different threat categories (this is very common in the IUCN system); this point needs to be clarified in the introduction and in paragraph 4.3.

- the last valid global assessment is from 2011 (not 2013).

- the sentence that "C. panormitana is absent in Greece" needs a reference, since the MS of Jiménez-Mejías et al. (2014) seems to refer to the Balkan Peninsula.

- the IUCN criteria are not from 2019, you are probably referring to the latest guidelines.

- if I understand correctly, translocations are not recommended; however then it is not clear why it is recommended to conserve seeds ex-situ and to multiply the species? wouldn't it be enough to protect the most threatened populations from disturbance?

- it is necessary to check the consistency between the bibliography cited in the text and the titles present in the bibliographic list.

Reviewer 1 ·

Basic reporting

In this manuscript Benítez-Benítez et al. are doing a comprehensive population genetic study with a group of closely related taxa of Carex, In addition, the authors are modelling their niche and projecting them to several scenarios of climatic change; apart from estimating lost/gained areas, they are doing a worthy attempt to see how the levels of genetic diversity and the genetic structure may change at the end of this century with respect to the present. Both the studied species (Carex, a well-known group with cold affinities) and the studied area (the Mediterranean Basin, whose climate will be among the most affected in the light of climate change) are particularly suited for this kind of approximation.
The methodology, both for the genetic and niche modeling aspects, is generally correct, with well-performed analyses. It is a pity, however, that the authors are still using the climatic scenarios of the Fifth Report of the IPCC (2014) and, thus, have not attempted to use the most updated climatic scenarios of the Sixth Report of the IPCC (CMIP6), that are available at least since early 2021. While the RCPs have been recalculated, they are combined with different socioeconomic assumption, i.e., the SSP or “Shared Socioeconomic Pathways”.
The Discussion is interesting and presented in a logical manner, though perhaps a little bit long (the conservation part can be easily reduced a little bit). In general, I liked the paper and I recommend publication in PeerJ.

Some minor questions:
(1) I suggest the authors to note that the estimation of ten genetic diversity and genetic structure under the scenario of climate change assumes no migration (i.e., that the populations outside the inferred potential distribution will be extinct and the populations inside will remain in the same place), which is an important limitation of this approximation (and should be acknowledged).
(2) The estimation of the ecological niche using the PCA for the future (Suppl. Fig. 4) is also assuming that the populations of the three species will remain in the same place as they occur today, which is quite unrealistic.
(3) Line 406: the authors use genetic data to infer the location of potential refugia, which is correct. The authors, however, could take advantage of the SDM and infer the location of climate refugia at the LGM and even for older periods. There are paleo-climatological data till the Pliocene in the PaleoClim database (http://www.paleoclim.org/) or in Oscillayers (https://onlinelibrary.wiley.com/doi/full/10.1111/geb.12979).

Experimental design

The experimental design is the usual in population genetics and ecological niche modelling. All used parameters and statistical tests are conventional and suitable for the present study.

Validity of the findings

No comment.

Reviewer 2 ·

Basic reporting

In the present manuscript, Benítez-Benítez and collaborators analysed the genetic structure of two sister species within the genus Carex, C. panormitana and C. reuteriana, using AFLP data. After disentangling the genetic structure within the complex, they studied the ecological niche of both species and subordinate taxa at present but also projected to the future under different climatic scenarios. They show how not only the putative favourable areas will be reduced in the future, but also how genetic diversity will be reduced as a consequence of the loss of populations.
The manuscript is well written with a careful English grammar, it is also well structured, with the introduction following the same structure than the discussion, stating clear objectives and hypotheses. Figures and tables are clear and clearly show the obtained results, and relevant literature on the field of study is cited.

Experimental design

The authors performed a great variety of analyses using different approaches, which led to the obtaining of strong results. For example, six different algorithms were used for the species distribution modelling, future projections were performed under another six different general circulation models, etc. Thus, I think the methods are well implemented, and despite the lower resolution of the AFLP when compared to high throughput sequencing techniques, the authors obtained good resolution to adress their questions.

Validity of the findings

Results are well supported by the strong and complete methodology followed by the authors. Even in the title, authors said that the study species are ecologically relevant, which also makes this research of interest for the understanding of those ecosistems and the consequences of the global climate change on their evolution in the future. In the text, the relevance of the species is justified by the abundance of these species in some ecosystems. My concern is whether these species, besides the abundance in riparian ecosystems, can play a role on those ecosystems, in sense of relationships with other species. After reading the manuscript and especially the threat of the species by overgrazing, I am thinking about this species as a primary resource for herbivorous species, and some other sources of importance for the ecosystems.
In this sense, I would add more ecologically information to the manuscript, not only to the introduction but also discussing the importance and implication that have the species on their ecosystems, and the consequences of their putative extinction. With a bit more ecological focus of the manuscript, this can be of interest for a broader audience and will generalized the scope of the research to other species and not only the two under study.

Additional comments

I found this manuscript is suitable for publication in PeerJ after some minor revisions and corrections as I specified above, and after incorporating the following:

Abstract:
Line 26 – Add comma after “…probably endangered”

Keywords:
Most keywords are already in the title or the abstract. I would suggest including some new keywords related to the manuscript, avoiding those in the title and/or abstract, for a more efficient indexing of the manuscript once published.

Introduction:
Lines 113-115 - The conservation will depends on the scope of the assessment (E.g. if regional or global). Thus, its conservation status is NOT disputed, but conditioned to the range considered for the assessment of the given species. If you wish to assess this species from a global perspective, maybe you can compare different (but global) red list.

Materials and Methods:
Line 153 – Specify version for PAUP
Line 169 – Specify version for GeneAlEx

Results:
Line 278 – “Carex” instead “C.” at the beginning of a sentence.
Line 306 – “Carex” instead “C.” at the beginning of a sentence.

Discussion:
Line 369 – Comma after “…subspecies reuteriana”
Line 394 – “North of Africa” instead “North Africa”
Line 397 – Remove “to” at the beginning of this line.
Line 451 – Rewrite this line as follows for a clearer comprehension: “endemism (Sicilian-Tunisian vs. Sardinian populations, Fig. 2) could remain stable over time.”
Line 523 – Correct this sentence to “…there are also at least…”
Lines 524-525 – Do you have some evidence for this species or another Carex species for the favouring of vegetative reproduction by overgrazing? Or is the overgrazing just preventing from sexual reproduction by herbivory of flowering shoots? if the latter, the rate of vegetative reproduction would remain stable and is the sexual reproduction rate what is decreasing. In this case, maybe you could rephrase this sentence.

---

## Round 0.2 · accepted · Accept

All the issues highlighted in the previous version have been resolved and the manuscript can be accepted in its current form.

Reviewer 1 ·

Basic reporting

The authors have correctly addressed my concerns. So, I think that the ms. is now in good shape.

Experimental design

No comment.

Validity of the findings

No comment.

Reviewer 2 ·

Basic reporting

In the previously-submitted version of the manuscript I commented some minor corrections that authors should be prior the acceptance of this paper. In this reviewed version, they incorporated most comments and justified why some (only few) other comments I made are not relevant to be included in this manuscript, in most cases due to lack of information about the study species/complex.

Then, I think this manuscript could be accepted for publication in PeerJ as is.

Experimental design

no comment

Validity of the findings

no comment